# SARS-CoV-2 and Adolescent Psychiatric Emergencies at the Tübingen University Hospital: Analyzing Trends, Diagnoses, and Contributing Factors

**DOI:** 10.3390/ijerph21020216

**Published:** 2024-02-12

**Authors:** Priska S. Schneider, Michelle Pantis, Christine Preiser, Daniela Hagmann, Gottfried M. Barth, Tobias J. Renner, Katharina Allgaier

**Affiliations:** 1Department of Child and Adolescent Psychiatry, Psychosomatics and Psychotherapy, University Hospital of Psychiatry and Psychotherapy, 72076 Tübingen, Germany; michelle.pantis@med.uni-tuebingen.de (M.P.); gottfried.barth@med.uni-tuebingen.de (G.M.B.); tobias.renner@med.uni-tuebingen.de (T.J.R.); katharina.allgaier@med.uni-tuebingen.de (K.A.); 2German Center for Mental Health (DZPG) Partner Site, 72076 Tübingen, Germany; 3Institute of Occupational and Social Medicine and Health Services Research, University Hospital, 72074 Tübingen, Germany; christine.preiser@med.uni-tuebingen.de; 4Centre for Public Health and Health Services Research, University Hospital Tübingen, 72016 Tübingen, Germany; 5LEAD Graduate School and Research Network, University Tübingen, 72072 Tübingen, Germany

**Keywords:** child and adolescent psychiatric emergencies, emergency admissions, COVID, SARS-CoV-2, eating disorders, obsessive–compulsive disorders, affective disorders, expansive disorders, anxiety disorders, substance abuse, psychoses, mixed-method, categories for psychiatric crisis, risk factors

## Abstract

Psychiatric emergencies have increased in recent decades, particularly with the onset of the SARS-CoV-2 pandemic, and far too little is known about the backgrounds of these emergencies. In this study, we investigated the extent to which the number of psychiatric emergencies changed during and in the aftermath of the SARS-CoV-2 pandemic at the Child and Adolescent Psychiatry (CAP) Tübingen. We considered age and psychiatric diagnoses. Additionally, we evaluated the backgrounds of the emergencies. We applied a mixed- (quantitative and qualitative) methods approach to data on emergency presentations at the CAP Tübingen from the pre-SARS-CoV-2 pandemic period (October 2019–January 2020) to Restriction Phase 1 (October 2020–January 2021), Restriction Phase 2 (October 2021–January 2022), and endemic phase (October 2022–January 2023). The total number of emergencies and those with eating disorders and affective disorders increased, while obsessive–compulsive disorders, expansive disorders and anxiety disorders decreased. The patients presenting in the pre-SARS-CoV-2 pandemic period were younger than those in the subsequent periods. We content-coded the reasons behind the emergency presentations. We also identified four areas of stressors and personality characteristics associated with the emergency presentations. In light of the increasing number of psychiatric emergencies, the long-term aim should be to meet the growing demands and create options for prevention.

## 1. Introduction

The SARS-CoV-2 pandemic has been an exceptional medical, social, and psychological situation with somatic, economic, and psychological effects. People around the world have experienced varying degrees of change and restrictions during the different phases of the pandemic. For children and adolescents, restrictions such as curfews and contact restrictions, especially school and kindergarten closures, have coincided with sensitive developmental phases [1]. Accordingly, it has been shown that the mental stress of children and adolescents has increased over the various phases of the SARS-CoV-2 pandemic [2,3].

In its most pronounced form, mental stress can result in the need for emergency care in Child and Adolescent Psychiatry (CAP). Psychiatric emergency presentations are exceptional mental situations associated with an acute or potential threat to life, and thus require urgent treatment [4,5]. The international literature and studies in the service area of the Department of Child and Adolescent Psychiatry, Psychosomatics and Psychotherapy at University Hospital Tübingen (CAP Tübingen) have shown that emergencies were already increasing in the decades before the SARS-CoV-2 pandemic [4,6], but an additional strong upward trend began during the SARS-CoV-2 pandemic [7,8,9]. Given that not much is known about the correlates and backgrounds of psychiatric emergency presentations in children and adolescents, with this study, we aimed to provide an update on the prevalence of such emergency presentations and to explore the underlying age groups, psychiatric diagnoses, and reasons for psychiatric emergency presentations in detail. We conducted an observational study with a mixed- (quantitative and qualitative) methods approach.

### 1.1. The SARS-CoV-2 Pandemic as a Social-Emotional Crisis for Children and Adolescents

The SARS-CoV-2 pandemic and corresponding restrictions (e.g., school closures, curfews) resulted in significant changes in the lives of children and adolescents [3,10,11]. The virus itself caused fear of contamination and worries about the health of oneself and others [3,11]. An extensive consumption of (social) media with heterogenous and conflicting information about SARS-CoV-2 exacerbated fears and worries [12].

To counteract the pandemic, restrictions were placed on social and school life. Such restrictions included school closures, mandatory quarantine periods, contact bans, the cancellation of private and public events, and requirements to wear masks. Many families remained in their homes for long periods of time [3,10,13]. These regulations had a broad impact on various social contexts of children and adolescents, including school, peers, and family.

Although some exceptions have been found, the overall impact of the pandemic on the mental health of children and adolescents has been found to be negative [1,3,14]. Beyond general well-being, the most commonly studied outcomes have been symptoms of depression and anxiety, such that the pandemic was found to intensify such symptoms [3,8]. Empirical studies have also shown that adolescents were more likely to suffer from increased symptoms than children [7]. This finding is in line with the idea that older adolescents are especially likely to face developmental challenges outside their homes. If not managed effectively, a particular vulnerability to psychological stress can result [7].

With regard to relevant contexts of children and adolescents, some studies have focused on the school context during and after the SARS-CoV-2 pandemic. Online teaching was associated with a drop in engagement and motivation for lessons. In addition, it was associated with sleep problems, stress, and mood problems in students (for a review, see [15]). Going back to school during the endemic phase when schools reopened posed another challenge for some children and adolescents [16]. So far there have been only a few studies on the effects of the SARS-CoV-2 pandemic in areas of adolescent life outside of school, but adolescents have reported the importance of social relationships with peers and the lack thereof during the Restriction Phases [1].

### 1.2. Child and Adolescent Psychiatric Emergencies

“A psychiatric emergency is an acute disturbance of behaviour, thought or mood of a patient which if untreated may lead to harm, either to the individual or to others in the environment” [5] (p. 59). As a psychiatric emergency can be a life-threatening crisis, specific indicators of risk can be described. Such indicators can be divided into endangerment to oneself and endangerment to others [17], with endangerment to oneself occurring more frequently (81.1%) than endangerment to others (22.6%). The most common risk characteristics are self-harm and suicidality [9,18,19]. Franzen et al. [18] found that the rate of emergency presentations with self-harm was 22.5%, whereas Wong et al. [9] reported it to be around 50% of all emergency presentations. Eichinger et al. [20] analyzed that suicidality was the main reason for presentation in over 50% of emergency presentations in Austria. These indicators of risk are the concrete causes of emergency presentations.

Even before the SARS-CoV-2 pandemic, a significant increase in emergency presentations had been observed internationally [21,22].

In Germany, child and adolescent psychiatric emergencies are treated in specialized child and adolescent psychiatric hospitals. The number of emergencies has increased in recent decades [23]. In line with this increase, the ratio of planned to emergency treatments shifted strongly toward emergency treatment [24,25,26]. Specifically, at CAP Tübingen, the rate of increase in emergencies was investigated for the time period from 1996 to 2014, and an emergency case increase of 405% was identified. When taking into account the number of patients instead of the number of emergencies, meaning that every patient was counted only once, the rate of increase was 354% (the reason for this difference is that the amount of patients with more than one emergency increased as well) [6].

The SARS-CoV-2 pandemic accelerated this increase. There is international evidence of an initial reduction with a subsequent increase during the early phases of the pandemic [19,21,27].

General information about changes in the numbers of emergencies in child and adolescent psychiatric hospitals in Germany during the pandemic is missing. However, it was evident that the already strained care system for child and adolescent mental health was further burdened by the rising number of emergencies. In some areas of Germany, the demand for psychological care had reached the limits of existing capacities [28].

For CAP Tübingen specifically, Allgaier et al. [7] compared emergency presentations at CAP Tübingen during Restriction Phase 1 of the SARS-CoV-2 pandemic (October 2020–January 2021) with the same time the year before (October 2019–January 2020). The results show an overall increase of 30% in emergency presentations, an increase of 33% in outpatient emergency visits (48 to 64), an increase of 28% in emergency admissions (72 to 92), and an increase of 29% in emergency telephone consultations (126 to 163). When considering each patient once, a similar pattern emerged, with one difference related to the overall increase in emergency presentations (10% in this analysis). This discrepancy could be attributed to the occurrence of multiple presentations by some patients, such as through telephone consultations and outpatient emergency visits. Additionally, the average age of patients increased during this period [7].

Previous studies conducted before the SARS-CoV-2 pandemic investigated which diagnoses were most frequent in children’s psychiatric emergency presentations. The most common diagnoses were depression, anxiety disorders, substance abuse disorders, personality disorders, and psychoses [18,29]. After the pandemic began, some studies focused on specific diagnoses or symptom areas with regard to their prevalence in emergencies during the pandemic.

One clear finding, reported in several studies on mental health crises in CAPs during the pandemic, was that eating disorders (EDs) increased [7,8,30].

Regarding obsessive–compulsive disorders (OCDs), Allgaier et al. [7] found a decrease in emergencies during the second wave of the SARS-CoV-2 pandemic (in comparison with the pre-SARS-CoV-2 pandemic period). Apart from emergencies, findings on OCDs during the SARS-CoV-2 pandemic have been heterogenous, but reviews have generally indicated that the symptoms of those already affected by OCDs worsened during the SARS-CoV-2 pandemic [31,32].

For the change in emergencies in CAPs for affective disorders during the pandemic, results have been heterogenous, ranging from no differences in comparison with pre-pandemic times in an earlier study [33] to a significant increase in a rather new study [34]. For expansive disorders, results have again been heterogenous, with studies finding a decrease in aggressive behavior [34] and a consistent appearance of agitation [33]. For disruptive, impulse control, and conduct disorders, Ferro et al. [34] did not find any differences over time, whereas Shankar et al. [35] found an increase. Anxiety disorders could cautiously be concluded to have declined in emergency presentations [34,35]. However, anxiety disorders seemed to increase during the pandemic in children and adolescents [36,37]. For the area of substance abuse disorders, studies have described a decrease [38], a consistent appearance [35,38,39], or an increase [34]. For psychoses, previous results have been mixed with studies identifying a consistent appearance [35] or a decline [40].

If we go beyond specific indicators of risk or diagnoses and look at the actual characteristics of emergencies, the literature becomes even more sparse. In one exception, an Austrian study from 2017 [20] looked at internal symptoms, external symptoms, and suicidality, but also added external factors as a fourth category to the classification scheme. This fourth category described problems in the family or at school or work. The study did not provide a more in-depth description of these external factors.

### 1.3. Developmental Areas of Children and Adolescents and Mental Illness

Various lines of research have conducted detailed examinations of the areas and topics that can be stressful for young people [41]. Such topics may be related to areas of development for children and adolescents [21,40]. Therefore, it is important to consider that, throughout their development, children and adolescents are asked to meet demands from the various social contexts they find themselves in [20,42]. When these demands are challenging, a lack of sufficient coping strategies can result in significant stress. Failure to cope with this stress can, under certain circumstances, lead to the development of mental illness or behavioral problems [42]. Steinhoff et al. [41] identified school, peers, intimate relationships, and family as social contexts in which stressful life events can occur. For example, in the school context, they identified repeating a grade as stressful, whereas in the peer context, violence and sexual victimization were identified as stressful. For intimate relationships, it was the loss of friends and separation from partners, and in the family context, it was loss and instability. Epstein et al. [43] found that academic stress factors, such as perfectionism or truancy, as well as social exclusion, were associated with a higher risk of self-harming behavior.

Mattejat et al. [44] categorized the relevant social contexts of mentally ill children and adolescents and their quality of life into six areas: school, family, friends, free time (interests and leisure activities), physical health, and mood (mental health). The Multidimensional Students’ Life Satisfaction Scale (MSLSS) also describes similar social contexts, namely, friends, family, school, self, and living environment [45].

Carballo et al. [46] identified some of these topics as risk factors for suicidal behavior. They found stressful life events in the context of family and peer conflicts to be a risk factor. The study showed that children’s and adolescents’ individual personality factors, including neuroticism and impulsivity, can be risk factors for suicidal behavior as well. Moreover, they found that family conflicts are also associated with suicidal behavior. Other risk factors detected by Carballo et al. [46] include a lack of family support, physical violence by a parent, unemployment, low income, neglect, divorce, loss of a parent, or violence in the family.

The studies described above [21,41] show relevant social contexts and personality characteristics that are associated with risk factors for mental illnesses. However, such categories have not yet been linked to child and adolescent psychiatric emergency presentations, so risk factors have yet to be identified in this regard.

Altogether, previous research has overwhelmingly shown how child and adolescent psychiatric emergencies have become more frequent in recent decades [6,22]. In particular, studies have shown that the beginning of the SARS-CoV-2 pandemic accelerated this process. There is initial evidence that there has been a change in the demographics of patients in the form of a shift toward older adolescents [7]. For some disorders, such as EDs, an exaggeration associated with the SARS-CoV-2 pandemic is evident [47]. For other disorders (e.g., OCDs, affective disorders, expansive disorders, substance abuse disorders, and psychoses), results have been heterogenous [31,32,33,34,35,38,39].

Far too little is known about the in-depth backgrounds of child and adolescent emergencies. Whereas indicators of risk, namely, endangerment to oneself and endangerment to others, have been identified [18], the factors that underlie the indicators are unknown. One reason for this lack of knowledge is that a system for categorizing the factors that underlie child and adolescent psychiatric emergencies has yet to be created.

The first goal of this paper was to update the study by Allgaier et al. [7] by including the time periods from October 2021 to January 2022 (Restriction Phase 2) and from October 2022 to January 2023 (reopening phase/endemic phase, hereafter referred to as the endemic phase). The time periods analyzed by Allgaier et al. [7] were also considered once again.

We addressed the following research questions:How did the number of emergency presentations, including outpatient emergency visits, inpatient emergency admissions, and telephone consultations, change between Restriction Phase 1 (October 2020–January 2021), Restriction Phase 2, and the endemic phase? We hypothesized that the numbers would increase in all categories over the time periods;How did the age of the patients change in the time between Restriction Phase 1, Restriction Phase 2, and the endemic phase? We hypothesized that there would be a shift toward more older adolescents;How did the respective number of patients with the following diagnoses change in the time period between Restriction Phase 1, Restriction Phase 2, and endemic phase?
3a.Eating disorders;3b.Obsessive–compulsive disorders.

Additionally, how did the respective number of patients with the following diagnoses change in the time period between pre-SARS-CoV-2 pandemic (October 2019–January 2020), Restriction Phase 1, Restriction Phase 2 and endemic phase?

3c.Affective disorders;3d.Expansive disorders;3e.Anxiety disorders;3f.Mental and behavioral disorders due to substance abuse over time;3g.Psychoses.

The next goal was to identify specific categories that described the reasons for emergency presentations and compare them between the different time periods (pre-SARS-CoV-2 pandemic, Restriction Phase 1, Restriction Phase 2, endemic phase). To do so, first, we developed a category system representing the reasons for emergency presentations (Research Question 4a), and then we analyzed the categories for changes over time (Research Question 4b).

## 2. Materials and Methods

### 2.1. Mixed-Methods Approach

We conducted an observational study using a mixed-methods approach that included both quantitative and qualitative analyses. This study was part of a larger project (Characteristics of Emergencies in CAP) approved by the institutional ethics committee at the University of Tuebingen. (848/2018BO1).

#### 2.1.1. Data Collection

The data originated from the university care center, the Department of Psychiatry, Psychosomatics, and Psychotherapy in Childhood and Adolescence, University Clinic for Psychiatry and Psychotherapy, Center for Mental Health, Tübingen (CAP Tübingen). At the time of the survey, the service area of the CAP Tübingen included the districts of Tübingen and Reutlingen as well as parts of the districts of Böblingen and Freudenstadt, which comprise both urban and rural regions. In addition to regular inpatient and outpatient treatment, emergency care is provided 7 days a week, 24 h a day. As part of this care, all emergency presentations are treated by professionally qualified staff.

Emergency presentations can be divided into three different types: outpatient emergency visits, inpatient emergency admissions, and emergency telephone consultations. Outpatient emergency visits are examinations of patients at the CAP Tübingen where, after a qualified consultation, it becomes apparent that the patients can be discharged home with appropriate counseling or initiation of necessary treatment steps. Inpatient emergency admissions are visits that lead to short-term admission to the crisis intervention unit. The purpose of immediate treatment in the crisis intervention unit is to provide intensive, often short-term stabilization of the patient to make it possible to create a plan for additional treatment or care. Telephone consultations often precede outpatient or inpatient emergency admissions. Sometimes, however, there is only a telephone consultation, as the concerns can be resolved over the phone or through a regular appointment outside of emergency care.

#### 2.1.2. Sample

The sample comprised patients who contacted or came to the CAP Tübingen for emergency care. Telephone consultations, outpatient emergency visits, and inpatient emergency admissions were considered. The *n* = 1543 cases we investigated were distributed across 733 patients because some patients presented several times (54.2%). The average age of patients was 14.95 (min = 4.71, max = 23.04, median = 15.40) years; 65.7% of the patients were female (i.e., sex assigned at birth). To address Questions 1 to 3g, all *n* = 1543 cases were included in the analyses. This is referred to as the presentation level because it considers all presentations no matter whether a patient had already presented earlier as an emergency. By contrast, for analyses on the case level, each person was considered only once per time period. More specifically, a patient’s first presentation (telephone consultation, outpatient emergency visit, or inpatient emergency admission) in one time period was considered. The resulting sample included *n* = 793 cases.

For Questions 4a and 4b, 77 patients were considered following a purposeful sampling strategy [48].

#### 2.1.3. Survey Period

In addition to the periods described in Allgaier et al. [6] from 1 October 2019 to 31 January 2020 (pre-SARS-CoV-2 pandemic) and from 1 October 2020 to 31 January 2021 (Restriction Phase 1), all emergency presentations in the periods from 1 October 2021 to 31 October 2022 (Restriction Phase 2) and from 1 October 2022 to 31 January 2023 (reopening phase/endemic phase, hereafter referred to as the endemic phase) were included in the analyses. These periods were chosen because they provided a particularly good contrast between the pre-SARS-CoV-2 period, the SARS-CoV-2 pandemic periods, and the endemic phase. The fall and winter months of 2020/2021 and 2021/2022, referred to as Restriction Phase 1 and Restriction Phase 2, were particularly affected by restrictions from the SARS-CoV-2 pandemic based on varying incidences of coronavirus infections. The associated life restrictions were implemented partly nationwide and partly specific to individual federal states (such as Baden-Württemberg, Tübingen). In the second measurement period, Restriction Phase 1, there were restrictions in schools, ranging from teaching only small groups of students to school closures. The restrictions in public life consisted of a partial curfew, a mandatory quarantine, and other strong restrictions on social life. At the third measurement point (Restriction Phase 2), the restrictions were somewhat more liberal, but similar. There were still restrictions on social life, mandatory quarantine periods, and at least mask requirements in schools. For the endemic phase, there were no more restrictions in schools and only very light restrictions on social life [13].

### 2.2. Quantitative Data Analysis

The quantitative analysis is an update and extension of Allgaier et al.’s [7] study.

#### 2.2.1. Measurement Instruments

We used electronic documentation of emergency visits, electronic patient records, and outpatient and inpatient medical reports in this study. Patients’ age (documentation and record), number of visits (documentation), and assigned diagnoses (record, medical report) were derived from these records.

#### 2.2.2. Evaluation of the Data

Analyses were performed using the statistical software R, version 4.3.2 [49]. To investigate differences in the ages of the patients between the four time periods, we compared the mean values with a one-way analysis of variance (ANOVA) [50]. For further analysis, patients were divided into three age groups (≤12.99 years; 13.00–15.99 years; ≥16.00 years) representing children/young teenagers, teenagers, and older teenagers/adolescents. We used a chi-square test of independence [51] to analyze differences in the distributions of people across the age groups between the four time periods. We used Fisher’s exact test [52] to analyze differences in the frequencies of the different disorders between the time points. This test was chosen due to the rather small number of cases.

### 2.3. Qualitative Data Analysis 

#### 2.3.1. Measurement Instruments

To address Questions 4a and 4b, we examined outpatient and inpatient medical reports.

Selection of data: To obtain a representative sample, the medical reports were selected on the basis of various criteria including age, gender, time of presentation, type of emergency presentation, number of emergency presentations, and treating medical personnel (see also Table 1). Outpatient emergency visits and inpatient emergency admissions without a medical report were excluded from the analyses, resulting in the exclusion of five presentations. Because no thorough corresponding medical reports were created for the telephone consultations, these consultations were also excluded. Besides the aforementioned criteria, the medical reports were chosen randomly. For patients with more than one emergency presentation, at least the first medical report from the first emergency presentation was considered. The numbers of medical reports per person and in general were determined by a content-related criterion: At the case level, we considered as many medical reports as necessary to achieve data saturation [53,54], which means that no new content could be generated by adding more medical reports. In total, 158 medical reports from 77 patients with outpatient emergency visits and inpatient emergency admissions were examined to analyze the two categories of emergency presentations in the CAP. This number corresponds to 10% of the patients.

#### 2.3.2. Analysis of the Data

We applied qualitative content analysis to categorize the content using a coding frame [55]. The qualitative content analysis involved concept-driven and data-driven categories, meaning that the categories were built from both the literature and the empirical data, respectively, during the coding process. The categories were further structured in accordance with Kuckartz and Rädiker [55].

The empirical data came from the medical reports. More specifically, every medical report included a “reason for presentation” section in which the specific indicators of risk, symptoms, stress factors, and contextual factors were recorded as described by the patients and caregivers. This section was used to form the data base.

Concept-driven categories of risk characteristics and contextual factors were derived from previous literature (a priori coding paradigm) [56]. Adapted from Steinhoff et al. [41], the stressful life events category was divided into the subcategories of school, peers, and family. Two independent coders (PSS and MP) examined 10 medical reports and assigned the specific stress factors to the three subcategories (i.e., school, peers, and family). An additional data-driven category (i.e., self) was added later. Via this procedure, we established whether the specified categories matched the data and whether the coding frame needed to be extended. The categories were then differentiated through a data-driven process. At a superordinate level, this process was effective for forming the main categories.

After we finalized the coding frame (see Appendix A Appendix A) and received feedback from a co-author (CP), two independent coders (PSS and MP) evaluated all the remaining data. The percentage agreement [55] between the coders was determined to be 98%. Disagreements were resolved through discussion between the team members (PSS and MP). For quality assurance, the categories were discussed with additional authors (CP and KA).

Furthermore, a quantitative analysis of the categories was conducted in the context of the examined time periods. The aim was to delve deeper into the quantitative data presented in the first part of the study and refine the multiperspective approach [57].

## 3. Results

### 3.1. Quantitative Results

#### 3.1.1. Differences in the Number of Emergency Presentations over Time

Table 2 and Table 3 show the results of the quantitative analysis of the development of emergency presentations across all four periods (from 1 October 2019 to 31 January 2020, from 1 October 2020 to 31 January 2021, from 1 October 2021 to 31 January 2022, and from 1 October 2022 to 31 January 2023). We counted the number of presentations at both the presentation level (each emergency presentation was counted) and the case level (only one emergency per patient per time period was counted), each separated according to the type of presentation for the respective emergency. We distinguished between outpatient emergency visits, inpatient emergency admissions, and emergency telephone consultations. When we measured the total volume, emergency presentations increased across the four time periods (*n* = 246 < 319 < 485 < 493 at the presentation level and *n* = 165 < 172 < 222 < 240 at the case level; Table 2 and Table 3). Emergency presentations also increased across all types of presentations.

**Table 2 ijerph-21-00216-t002:** Comparisons of frequency by age group for patients with different types of emergency presentations in all periods on the presentation level.

Kind of Emergency Presentation	Time Period	*N* Total and by Age Group ^a^						Mean Age Differences between the Groups							
		All	≤12.99 Years	13.00–15.99 Years	≥16.00 Years	χ^2^(2)	*p*	M	SD	Min	Max	F	df	*p*	^c^ η^2^
All	2019/2020	246	66 (27%)	93 (38%)	87 (35%)	37.48	<0.001 *	14.32	2.70	7.12	18.32	15.02	1540	<0.001 *	<0.01
	2020/2021	319	48 (15%)	153 (48%)	118 (37%)			14.99	2.03	6.67	18.00				
	2021/2022	485 ^b^	57 (12%)	249 (51%)	178 (37%)			15.06	2.07	6.90	18.21				
	2022/2023	493	67 (14%)	214 (43%)	212 (43%)			15.10	2.41	4.71	18.65				
Telephone	2019/2020	126	43 (34%)	39 (31%)	44 (35%)	28.82	<0.001 *	14.06	2.97	7.12	18.32	8.71	802	0.005 *	0.01
	2020/2021	163	29 (18%)	80 (49%)	54 (33%)			14.78	2.17	6.67	17.99				
	2021/2022	257 ^b^	38 (15%)	126 (49%)	92 (36%)			14.95	2.18	6.90	18.21				
	2022/2023	259	41 (16%)	111 (43%)	107 (41%)			14.93	2.65	4.71	18.65				
Outpatient	2019/2020	48	15 (31%)	19 (40%)	14 (29%)	11.21	0.082	13.83	2.91	7.15	17.83	1.84	309	0.020 *	<0.01
	2020/2021	64	8 (13%)	27 (42%)	29 (45%)			15.23	1.89	9.39	17.99				
	2021/2022	104	13 (12%)	53(51%)	38 (37%)			14.84	2.22	8.63	17.99				
	2022/2023	95	17 (18%)	40 (42%)	38 (40%)			14.77	2.49	6.22	17.84				
Admission	2019/2020	72	8 (11%)	35 (49%)	29 (40%)	8.30	0.217	15.10	1.77	9.68	17.84	6.35	425	0.090	0.01
	2020/2021	92	11 (12%)	46 (50%)	35 (38%)			15.20	1.83	9.92	18.00				
	2021/2022	124	6 (5%)	70 (56%)	48 (39%)			15.49	1.60	10.54	18.00				
	2022/2023	139	9 (7%)	63 (45%)	67 (48%)			16.64	1.73	6.22	18.01				

Note. χ^2^(2) and *p* refer to chi-square independence tests of the distributions of individuals across the age groups. F, df, *p*, and η^2^ refer to ANOVAs for age group comparisons. All time periods were compared for each type of presentation. η^2^ denotes the effect size eta-squared. Each patient’s presentation was included in the data on an equal footing. Telephone refers to telephone consultations, Outpatient refers to outpatient emergency visits and Admission refers to inpatient emergency admission. ^a^ The percentages refer to the total number of children and adolescents with the type of presentation and period described in the same line. ^b^ There was one missing value for age. ^c^ Reference values for eta-squared according to Ellis [58] were: 0.01 ≤ η^2^ < 0.06 small; 0.06 ≤ η^2^ < 0.14 medium; 0.14 ≤ η^2^ large. * describes a significant result at a significance level of α = 0.05.

#### 3.1.2. Differences in Patients’ Ages in Emergency Presentations over Time

We examined whether patients differed in age across the time periods (Table 2 and Table 3). We found significant differences in age across time periods at the presentation level (*p* <0.001, η^2^ < 0.01). A closer look at the differences revealed that patients with outpatient emergency visits or inpatient emergency admissions were significantly younger (*M* = 14.32 years) during the pre-SARS-CoV-2 pandemic period than these types of patients in the three subsequent periods (*M* = (14.99; 15.10); *p* = (0.003; <0.001)). Patients appeared to get older across Restriction Phase 1, Restriction Phase 2, and the endemic phase (*M* = 14.99 < 15.06 < 15.10 years), but this increase was not statistically significant (*p* = (0.912; 0.996)). Additionally, we found that the difference in age was statistically significant for outpatient emergency visits (*p* = 0.020) between the pre-SARS-CoV-2 pandemic period and Restriction Phase 1 (*p* = 0.011, *d* = 0.59), as well as for telephone consultations (*p* = 0.005) between the pre-SARS-CoV-2 pandemic period and Restriction Phase 2 (*p* = 0.005, *d* = 0.36), and between the pre-SARS-CoV-2 pandemic period and the endemic phase (*p* = 0.007, *d* = 0.32).

Results for age were similar at the case level (Table 3). The ANOVA showed an overall significant result (*p* = 0.004, η^2^ = 0.01). The significant increase in age for outpatient emergency visits between the pre-SARS-CoV-2 pandemic period (*M* = 13.83 years) and Restriction Phase 1 (*M* = 15.13 years) (*p* = 0.038) contributed to this result. Additionally, there was a significant increase in age between the pre-SARS-CoV-2 pandemic period (*M* = 14.12 years) and Restriction Phase 2 (*M* = 15.01 years) (*p* = 0.004), as well as between the pre-SARS-CoV-2 pandemic period and the endemic phase (*M* = 14.92 years) (*p* = 0.012).

Also, significant differences were found in how the patients were distributed across age groups over time (*p* < 0.001) at both the presentation and case levels (Table 2 and Table 3). Overall, there were significant differences across the time periods for the age group ≤ 12.99 years (*p* < 0.001) and the 13–15-year-olds (*p* = 0.003), but not for the age group ≥ 16.00 years. The post hoc analysis revealed significant differences between the pre-SARS-CoV-2 pandemic period and Restriction Phase 2 (*p* = 0.003) for 13–15-year-olds. Significant differences between the pre-SARS-CoV-2 pandemic period and Restriction Phase 1 (*p* < 0.001), Restriction Phase 2 (*p* < 0.001), and the endemic phase (*p*< 0.001) were found for the under 13-year-olds. There were significant differences in outpatient emergency visits for the under 13-year-olds (*p* = 0.025). For telephone consultations, there were significant differences (*p* < 0.001) across the time periods for the under 13-year-olds (*p* < 0.001) and 13–15-year-olds (*p* = 0.004).

#### 3.1.3. Differences in the Number of EDs over Time

Within EDs, at the case level, the diagnosis F50.0 (anorexia nervosa) occurred most frequently (69 occurrences). This was followed by F50.1 (atypical anorexia nervosa; 31 occurrences), F50.2 (bulimia nervosa; 14 occurrences), F50.3 (atypical bulimia nervosa; 10 occurrences), and F50.9 (eating disorder, unspecified; 17 occurrences).

An increase in presentations with an ED diagnosis (Table 4) was observed (*p* < 0.001, *Cramer’s V* = 0.09). Post hoc analyses showed that this change was significant when the pre-SARS-CoV-2 pandemic period (*n* = 5) was compared with Restriction Phase 1 (*n* = 24, *p* = 0.007), Restriction Phase 2 (*n* = 61, *p* < 0.001), and the endemic phase (*n* = 52, *p* < 0.001). When considering the change in individual types of presentations, only telephone consultations showed a significant change (*p* < 0.001, *Cramer’s V* = 0.1). Post hoc analyses showed significant differences when the pre-SARS-CoV-2 pandemic period was compared with Restriction Phase 2 (*p* < 0.001) and the endemic phase (*p* = 0.004). The increases in the two in-person emergency department presentations (Table 4) were not significant (*p* = 0.108).

An increase in patients with an ED diagnosis up to Restriction Phase 2 was observed for outpatient emergency visits and telephone consultations, with a slight decrease again in the endemic phase (Table 4). Fisher’s test provided a significant result for the change in the number of ED diagnoses across the periods at the case level (*p* = 0.003, *Cramer’s V* = 0.08). Post hoc analyses showed that the number of ED diagnoses differed when the pre-SARS-CoV-2 pandemic period (*n* = 3) was compared with Restriction Phase 1 (*n* = 14, *p* = 0.023), Restriction Phase 2 (*n* = 26, *p* = 0.003), and the endemic phase (*n* = 22, *p* = 0.008). Significant results were found only for outpatient emergency visits (*p* = 0.035, *Cramer’s V* = 0.13) and telephone consultations (*p* = 0.031, *Cramer’s V* = 0.09). A post hoc analysis revealed that the difference in the number of diagnoses in telephone consultations occurred primarily between the pre-SARS-CoV-2 pandemic period and Restriction Phase 2 (*p* = 0.035). For the two in-person emergency department presentations, no clear trend was identified.

#### 3.1.4. Differences in the Number of OCDs over Time

Within OCDs, at the case level, the diagnosis F42.2 (mixed obsessive–compulsive thoughts and actions) occurred most frequently (27 occurrences). This was followed by F42.1 (predominantly compulsive actions; 9 occurrences) and F42.0 (predominantly obsessive thoughts or ruminative compulsion; 2 occurrences).

The temporal analysis of OCDs showed a decrease in the number of presentations of patients with an OCD diagnosis between the pre-SARS-CoV-2 pandemic period and Restriction Phase 2, and a sharp increase in the endemic phase (Table 4; *p* < 0.001, *Cramer’s V* = 0.08). There were also more OCD diagnoses in the pre-SARS-CoV-2 pandemic period (*n* = 9) and in the endemic phase (*n =* 22) than in the Restriction Phases (1, *n* = 2 and 2, *n* = 5) (Table 4). This observation was confirmed by significant differences between the pre-SARS-CoV-2 pandemic period when compared with Restriction Phase 1 (*p* = 0.026) and Restriction Phase 2 (*p* = 0.027). The difference in the number of diagnoses between the time periods was significant only for telephone consultations (*p =* 0.010, *Cramer’s V* = 0.08) but not for outpatient emergency visits or inpatient emergency admissions. The trend toward increased presentations in the pre-SARS-CoV-2 pandemic period and the SARS-CoV-2 pandemic period could not be observed at the case level (Table 4).

#### 3.1.5. Differences in the Number of Affective Disorders over Time

Within affective disorders, at the case level, the diagnosis F32.- (depressive episode) occurred most frequently (799 occurrences). This was followed by F33.- (recurrent depressive disorder; 8 occurrences), F34.- (persistent affective disorders; 4 occurrences), and F31.- (bipolar affective disorder; 2 occurrences).

Descriptively, there was an overall increase in affective disorder diagnoses up to Restriction Phase 2 (except for inpatient emergency admissions at the case level) and a decrease in cases in the endemic phase (except for inpatient emergency admissions and telephone consultations at the case level) (Table 4. Presentation level: *p* = 0.019, *Cramer’s V* = 0.06. Case level: *p* = 0.033, *Cramer’s V* = 0.08). At the presentation level, a post hoc analysis identified a significant increase in the number of presentations with an affective disorder between Restriction Phase 1 (*n* = 147) and Restriction Phase 2 (*n* = 280; *p* = 0.048). Otherwise, no additional significant effects were found.

#### 3.1.6. Differences in the Number of Expansive Disorders over Time

Within expansive disorders, at the case level, the diagnosis F90.0- (disturbance of activity and attention) occurred most frequently (97 occurrences). This was followed by F90.1 (hyperkinetic conduct disorder; 52 occurrences), F91.3 (oppositional defiant disorder; 37 occurrences), F91.2 (socialized conduct disorder; 23 occurrences), and F91.0 (conduct disorder confined to the family context; 16 occurrences).

Decreases in presentations and cases with an expansive disorder diagnosis were observed over time for all emergency presentations (Table 4), specifically for telephone consultations (*p* < 0.001, *Cramer’s V* = 0.12 at the presentation level and *Cramer’s V* = 0.14 at the case level) and outpatient emergency visits. There was a slight increase in presentations and cases in the endemic phase for most types of presentations. The changes in the numbers of presentations and cases for all types of presentations with an expansive disorder diagnosis were statistically significantly different over time (*p* < 0.001, *Cramer’s V* = 0.11). This finding can be explained by changes in the number of diagnoses between the pre-SARS-CoV-2 pandemic period (*n* = 55 at the presentation level; *n* = 34 at the case level) and Restriction Phase 2 (*n* = 54, *p* < 0.001 at the presentation level; *n* = 22, *p* = 0.005 at the case level), as well as the endemic phase (*n* = 47, *p* < 0.001 at the presentation level; *n* = 26, *p* = 0.011 at the case level), and between Restriction Phase 1 (*n* = 55 at the presentation level; *n* = 34 at the case level) and Restriction Phase 2 (*n* = 54, *p* = 0.002 at the presentation level; *n* = 22, *p* = 0.013 at the case level), as well as the endemic phase (*n* = 47, *p* < 0.001 at the presentation level; *n* = 26, *p* = 0.042 at the case level). Similar differences were found in telephone consultations (0.001 > *p* < 0.04), except for the difference between Restriction Phase 1 (*n* = 20) and the endemic phase (*n* = 14) at the case level. For inpatient emergency admissions (*p* = 0.005, *Cramer’s V* = 0.12), there was only a significant difference between Restriction Phase 1 (*n* = 19) and Restriction Phase 2 (*n* = 8, *p* = 0.018) at the presentation level.

#### 3.1.7. Differences in the Number of Anxiety Disorders over Time

Within anxiety disorders, at the case level, the diagnosis F40.1 (social phobias) occurred most frequently (67 occurrences). This was followed by F41.9 (anxiety disorder, unspecified; 34 occurrences), F41.1 (generalized anxiety disorder; 23 occurrences), F41.2 (mixed anxiety and depressive disorder; 18 occurrences), and F40.2 (specific (isolated) phobias; 16 occurrences).

At both the presentation and case levels, a descriptively observable but nonsignificant increase in cases with an anxiety disorder could be observed over the course of the pandemic (Table 5).

#### 3.1.8. Differences in the Number of Mental and Behavioral Disorders Due to Substance Abuse over Time

Within mental and behavioral disorders due to substance abuse, at the case level, the diagnosis F19.- (mental and behavioral disorders due to multiple drug use and use of other psychoactive substances) occurred most frequently (54 occurrences). This was followed by F10.- (mental and behavioral disorders due to use of alcohol; 27 occurrences), F12.- (mental and behavioral disorders due to use of cannabinoids; 18 occurrences), F17.- (mental and behavioral disorders due to use of tobacco; 12 occurrences), F13.- (mental and behavioral disorders due to use of sedatives or hypnotics; 11 occurrences), and F11.- (mental and behavioral disorders due to use of opioids) and F15.- (mental and behavioral disorders due to use of other stimulants, including caffeine), each with 7 occurrences.

The numbers of presentations with substance abuse barely differed in relative terms between the different types of presentations and the different time periods. At the case level, a decrease in cases with substance abuse beginning in Restriction Phase 2 was found when all types of presentations were considered (*p* = 0.025, *Cramer’s V* = 0.08). Post hoc analyses were nonsignificant (Table 5).

#### 3.1.9. Differences in the Number of Psychoses over Time

Within psychoses, at the case level, the diagnosis F23.- (acute and transient psychotic disorders) occurred most frequently (14 occurrences). This was followed by F20.- (schizophrenia; 10 occurrences), F22.- (persistent delusional disorders; 8 occurrences), F24 (induced delusional disorder; 2 occurrences), and F29 (unspecified nonorganic psychosis; 2 occurrences).

The number of presentations and cases with a diagnosis of psychosis remained rather constant across the time periods (Table 5). Only a descriptively observable but nonsignificant decrease in inpatient emergency admissions in patients with a diagnosis of psychosis was observed from pre-SARS-CoV-2 (*n* = 1) and Restriction Phase 1 (*n* = 2) to Restriction Phase 2 (*n* = 0) and the endemic phase (Table 5; *n* = 0).

### 3.2. Qualitative Results

The goal of the qualitative analysis was to identify specific categories of characteristics that were associated with emergency presentations (research question 4a). Through the deductive and inductive coding process, we formed a category. The main categories we developed were (1) specific reason for presentation, (2) stressors, and (3) personality characteristics. Furthermore, (only) the category (2) stressors was differentiated into subcategories (2a) school, (2b) peers, (2c) family, and (2d) self.

The subcategories resulted from a closer examination of the descriptions of the stressors and on the basis of the existing literature [46]. It became evident that the stressors could be meaningfully differentiated on the basis of the life contexts in which the development of children and adolescents takes place. All specific topics mentioned in the medical records could be assigned to one of the categories or subcategories (Table 6).

The categories are defined below.

(1)Specific reasons for presentation—The reason for the presentation consisted of the acute and usually specific cause of a child’s or adolescent’s emergency outpatient emergency visit or inpatient emergency admission;(2)Stressors—Stress factors consisted of burdens, issues, situations, circumstances, events, or conditions described by the patient themselves or a significant other. Such stressors were reported beyond the specific reason for presentation or admission. Upon closer examination of the described stressors and considering the existing literature [46], the stressors were divided into subcategories, specifically describing social contexts that represent the developmental areas of children and adolescents. It became evident that further differentiation into various life domains was possible. The categories that emerged as meaningful were school, peers, family, and self. We confirmed that the stress factors could be assigned to different life contexts in which the development of children and adolescents takes place:
(a)School—Social context within the learning environment where children and adolescents spend a significant portion of their daily lives;(b)Peers—Social context with flexible conditions, within which children and adolescents move to varying extents, but which play a significant role in their development. Peers typically refer to individuals of the same age with whom individuals have relationships;(c)Family—Social context in which children and adolescents grow up and must navigate through in their daily lives. Various relationship groups exist, with a focus on the central living place of children and adolescents;(d)Self—Biographical experiences, topics, and symptom areas that are individually described by the patient and caregivers as a current stress factor [59] and not directly related to a social context (a–c).
(3)Personality characteristics—Other than patient’s biographical experience or symptoms, this category includes the patient’s personality. Personality characteristics, by definition, represent relatively stable characteristics of individuals. They persist over time and in situations and generally have an influence on experiences and behavior [59,60].

To identify various connections between the existing categories and time periods (Research Question 4b), the aim was to further analyze any changes that occurred between the corresponding time periods. Figure 1, Figure 2 and Figure 3 and Appendix A Appendix A show the results for the main categories and subcategories, as well as the identification of specific recurring factors. At the level of specific reasons for presentation, we found that, at later time points, patients presented more frequently with suicidal behavior, whereas self-harm remained relatively stable. Risk to others appears to have been a particularly frequent reason for presentation during the pre-SARS-CoV-2 pandemic period. Absenteeism was a reason for an emergency presentation only from Restriction Phase 2 onwards. Pathological use of media was mentioned one time in the pre-SARS-CoV-2 pandemic period, but not in the subsequent periods. ED symptoms and being underweight as reasons for presentation were mainly found in the pre-SARS-CoV-2 pandemic period and during Restriction Phase 1, whereas ED symptoms continued to be found during the Restriction Phase 2 period.

At the level of personality characteristics, impulsiveness was found in all periods. High performance expectations were found only in the pre-SARS-CoV-2 pandemic period and the endemic phase. Perfectionism was found only in the pre-SARS-CoV-2 pandemic period. Self-doubt, on the other hand, was found in all periods except the pre-SARS-CoV-2 pandemic period.

Upon examining stress factors within the four life domains (school, peer, family, self) across different periods, we found that family was the most frequently mentioned stressor throughout. Notably, stressors were generally mentioned more frequently in all life domains during the pre-SARS-CoV-2 pandemic period and the endemic phase, whereas fewer stressors were mentioned during the two Restriction Phases. The difference was particularly noticeable in the self and family domains. In the endemic phase, the number of stress factors in the peer domain was much higher.

Regarding the stressor of family, conflicts with a family member or caregiver were frequently reported across all time periods. The stress caused by out-of-home placement (e.g., a group home) increased from Restriction Phase 1 onwards. Parental conflicts and parental separation were described as particularly high during Restriction Phase 1. However, there was only a slight increase in the mention of parental conflicts during Restriction Phase 1. The mention of violent confrontations as a stressor decreased over time. No clear trend over time was identified for the areas of death, financial difficulties, history of abuse, or lack of understanding toward the children.

Suicidal tendencies and self-harm as passive stressors tended to decrease across the time periods. Substance abuse, physical symptoms, and history of abuse appeared to be recurring stressors. The SARS-CoV-2 pandemic as a stressor was only directly mentioned once in Restriction Phase 1. Stress related to sexuality was mentioned only from Restriction Phase 1 onwards. For all other stressors in the area of self, no differences over time were found.

School absenteeism as a stressor increased in the endemic phase as the schools reopened. Academic stress and performance pressure were mentioned as stressors in all periods, whereas a decline in performance was reported exclusively during the pre-SARS-CoV-2 pandemic period. Concentration difficulties were also reported more frequently during the pre-SARS-CoV-2 pandemic period. Conflicts with classmates were mentioned as stressors in all periods, but less frequently during the school closures (Restriction Phase 1 and Restriction Phase 2). Bullying, on the other hand, was named as a stressor during the Restriction Phases, albeit less frequently during the endemic phase. School closures due to SARS-CoV-2 were exclusively mentioned as a stressor during the endemic phase.

During the endemic phase, peers were mentioned as a stress factor almost four times more frequently than in the preceding periods.

## 4. Discussion

The SARS-CoV-2 pandemic and its aftermath have significantly impacted the mental health of children and adolescents, resulting in an increase in psychiatric diseases. Emergency presentations in child and adolescent psychiatric facilities due to acute psychiatric crises were already increasing in recent decades, but these increases were even further exacerbated by the SARS-CoV-2 pandemic [6,7,8]. However, little is known about how underlying factors lead to acute crises in children and adolescents, although knowledge about such mechanisms is key for designing effective interventions for prevention and early intervention. By applying a mixed methods approach in an observational study, we examined quantitative data on emergency cases during the later parts of the SARS-CoV-2 pandemic and the onset of the endemic phase, as well as the relevant characteristics of the emergency cases. This work provides an update of a previous study by our group [7] and addresses additional aspects in an effort to elucidate critical mechanisms in the genesis of acute psychiatric crises in children and adolescents.

The quantitative evaluation showed a consistent increase in the number of emergencies from the pre-SARS-CoV-2 pandemic period to the endemic phase. This finding applied to both the number of presentations and the number of patients. Our results are consistent with previous research on the utilization of emergency care [6,7,8,21,22].

Regarding the ages of the patients presenting with emergencies during the time periods we analyzed, we found that the patients tended to be older than in previous non-pandemic periods. Complementing this finding, the numbers of emergency presentations in the youngest age group (under 13 years) decreased. Overall, we found that the differences in the proportions of patients from the different age groups were especially prominent in emergency telephone consultations.

In the SARS-CoV-2 pandemic period (Restriction Phases 1 and 2) and the endemic phase, the outpatient emergency visits or inpatient emergency admissions comprised mainly adolescent patients (aged 13 and over) rather than younger children. This overrepresentation of adolescents might be explained by the fact that the pandemic-related restrictions seemed to have affected adolescents differently compared to younger children. On the one hand, restrictions on social activities outside the family, including peer activities or sports, tend to have a more significant impact on the mental well-being of adolescents than of children [1,8]. Furthermore, the Restriction Phases particularly affected middle schools rather than elementary or lower schools [15]. The pandemic-related restrictions on private life, schooling, and general social activities placed enormous pressure on children and adolescents [10,11]. Combined with the intensive challenges that arise during the vulnerable phase of adolescence and are associated with a higher risk of mental health issues and depression [8], the pandemic-related restrictions clearly led to a significant increase in the need for psychiatric emergency care in adolescents. Our data on emergency presentations substantiate the vulnerability of adolescents’ mental health and underscore the urgent need for interventions and support services that can help prevent psychiatric disorders, especially for this age group.

Regarding EDs, increases in patients and emergency presentations were observed. This finding corresponds perfectly with previous data that showed a substantial increase in EDs during the SARS-CoV-2 pandemic [7,8,30]. It was especially striking in our data that there were more inpatient emergency admissions than outpatient emergency visits, indicating the acute severity of the disorder in the majority of the patients. Especially for EDs, early detection is improved by an active interplay between families, caregivers in the health system, and schools as one example of a key psychosocial environment for adolescents. During the Restriction Phases, schools’ early-warning functions were disabled, potentially contributing to the fact that patients presented with severe symptoms only in an emergency setting.

With regard to OCDs, we found a descriptive decrease in patients in the Restriction Phases. However, we observed a clear increase in the endemic phase. These results add relevant information to the heterogeneous literature on OCDs and the SARS-CoV-2 pandemic [7,31,32]. It might be assumed that the steps taken to counter the pandemic led to a sense of security among people with OCDs, but that this sense of security dissipated when the endemic phase began. Such a trend might explain the increase in crises that began immediately with the endemic phase.

The number of cases of affective disorders, which were mostly depressive episodes in the CAP Tübingen, increased during the pandemic, but dropped a little in the endemic phase. This result is in line with the increase in affective disorders during the pandemic found by Ferreo et al. [34]. The increase in affective disorders can be explained by the loss of social support due to the canceling of activities and growing stress in response to the pandemic. In general, it must be assumed that individual problems were reinforced by the pandemic and its restrictions, thus leading to an emergency deterioration of affective problems [33,34].

For expansive disorders, which mostly consisted of Attention Deficit Disorders and Attention Deficit Hyperactivity Disorders (AD(H)Ds) in our data set, followed by oppositional disorders, we found a decrease in emergencies during the Restriction Phases, a finding that contributes to the heterogeneous results found in previous studies [33,34,35,38]. Flik et al. [61] examined the change in AD(H)D diagnoses over time in the same sample, and also found a nonsignificant decline in AD(H)D during the Restriction Phases. Therefore, it can be assumed that during the Restriction Phases, children and adolescents suffering from AD(H)D may have experienced relief from school-related stress [34]. The differences in oppositional disorders between the pre/post-SARS-CoV-2 periods and the Restriction Phases should be examined in more detail in future studies.

For anxiety disorders, we did not find any significant differences across the time periods. But given that the trend in our data indicates an increase in anxiety disorders during the pandemic, the previously identified decline in anxiety disorders [34,35] could not be confirmed in our data.

In the area of substance abuse, we found a relatively stable pattern over time, which was in line with previous studies [35,39]. However, beginning in Restriction Phase 2, which was the lighter of the two restriction periods in our study [13], we found a decrease in the number of patients abusing substances, a finding that may represent a positive reaction to social life starting up again [13].

For psychoses, in line with Shankar et al. [35], we did not find any significant changes in the number of presentations or patients over time. Therefore, it is reasonable to assume that the pandemic did not have a specific, direct impact on the development of psychosis described in Ferrando et al. [40]. However, we had only a small sample in our study, thereby limiting the generalizability of our findings.

In our qualitative analysis of emergency presentations, on the basis of the existing literature on risk and stress factors, we extracted the reasons for the emergency presentations from the medical reports and categorized them into three main levels. These categories provided a nuanced understanding of the primary causes of the emergency presentations, along with stressors and individual personality characteristics as secondary reasons. In accordance with Eschenbeck et al. [42], we identified the school situation, peers, family factors, and, in a deductive process, self as subcategories of relevant stress events.

The medical reports described the specific reasons for the presentation, which typically represented an acute risk and consequently led to an inpatient admission or outpatient visit. Upon closer examination, however, additional themes were described. Thus, it became evident that an initial level of differentiation was necessary and that it would be beneficial to distinguish between the specific primary reasons for the presentations and the secondary themes, which we categorized into stressors and personality characteristics for the analyses.

Furthermore, a previous study found that social context factors significantly affect the internal and external factors that prompt acute presentations to clinics [20]. The findings on social context factors from the present qualitative analysis of outpatient emergency visits and inpatient emergency admissions at CAP Tübingen were in line with the ones identified in the literature. In this specific case, the analyses have confirmed that the previous findings held for emergency presentations at the CAP Tübingen as well. We were able to examine the subcategories of stressors (i.e., school, family, peers, and self) across the time periods as well. Our data confirm that the subcategories of stressful life events (i.e., school, family, peers, and self) were appropriate [42].

This observation thus confirmed that various stressors, situations, and contexts typically play a central role in mental disorders. In the final category, we considered personality characteristics as relatively stable risk factors for emergency presentations [60]. With these additional categories, the study enhances the understanding of mental health emergencies, and also contributes to the theoretical framework with respect to their categorization, which can now be used in other studies.

In the analysis of the time periods, certain categories were represented more frequently than others. Looking more closely into the specific reasons for presentations, suicidality emerged as the most common reason for emergency presentations, followed by self-injury and danger to others [20]. Suicidal tendencies also increased across the time periods included in our study. This trend is consistent with findings from previous literature [20,23]. Thus, there is consistent evidence that the SARS-CoV-2 pandemic affected the mental health of children and adolescents, especially by intensifying symptoms.

An examination of personality characteristics revealed that perfectionism and a demand for high performance, both of which are closely linked to the school context, were observed in our study, predominantly during the pre-SARS-CoV-2 pandemic phase and the endemic phase.

It can be assumed that these personality characteristics are closely related to school demands and performance assessments. When we considered the subcategories of school, peers, family, and self within the main category of stressors, school-related stressors (e.g., academic stress and performance pressure) were reported consistently across all the time periods, whereas a decline in performance was mentioned exclusively in the pre-SARS-CoV-2 pandemic phase. Additionally, difficulties concentrating were frequently reported during the pre-SARS-CoV-2 pandemic period. School absenteeism increased in the endemic phase. The topic of school closures was rarely mentioned, thus suggesting that the closures themselves might not have been perceived as directly stressful for students. However, there was a simultaneous increase in emergency presentations among teenagers during this period.

The re-entry into school during the endemic phase seemed to be perceived as a burden, as stressors in this regard were mentioned more frequently, and school absenteeism increased. The reopening of schools after a period of closure represented a significant change for children and adolescents [16]. When considering new requirements and social interactions, the transition from isolation during the Restriction Phases to a return to the school routine was challenging.

In general, there were no remarkable increases in how often the SARS-CoV-2 pandemic itself was mentioned as a specific stressor. Here, it can be assumed that children and adolescents were affected less by the SARS-CoV-2 pandemic itself, but more by its consequences and the steps that were taken to contain it.

Another noteworthy aspect is that the pathological use of media was mentioned once in the pre-SARS-CoV-2 pandemic period but not subsequently, and not as a stressor. The role of media use in the pandemic was discussed (e.g., in connection with the development of EDs) [8]. Our finding is in line with the study by Laczkovics [14], who found no connection between media consumption and psychopathology in a sample of adolescents (14–18 years) and in contrast to a sample of young adults (19–25 years). Altogether, this topic appears to be complex and should be pursued in future studies.

Considering the differences between the pre-SARS-CoV-2 pandemic period, the Restriction Phases, and the endemic phase, the qualitative analysis also revealed that peers were mentioned more frequently as stressors in the pre-SARS-CoV-2 pandemic period and even more so in the endemic phase. Because adolescents had been socially isolated during the Restriction Phases and were out of practice in interacting with peers, they were more likely to identify social interactions with peers as stressful events when the schools reopened.

Notably, Restriction Phase 1 was marked by a substantial increase in parental conflicts and instances of parental separation. However, conflicts with parents and violent confrontations did not exhibit a marked surge during the Restriction Phases. These findings suggest that while family-related stressors (e.g., conflicts and parental conflicts) were prevalent, the severity of conflicts, especially violent ones, did not increase substantially during the Restriction Phases. A number of unreported cases cannot be ruled out.

In summary, the qualitative analysis revealed a general decrease in stressors during the Restriction Phases, with minimal mention of SARS-CoV-2 itself as a stressor. While it was acknowledged as a factor, it was not the only contributor that was mentioned. The long-term implications of this trend, whether it was tied to the SARS-CoV-2 pandemic or persisted into the current endemic phase, require further investigation [47].

Our data do not indicate that single factors were responsible for the emergency presentations. It is possible to identify a variety of specific reasons for an emergency presentation, as well as various personality characteristics that described the emergency ideation in question. The need to differentiate between the specific reason for the emergency presentation and additional stressors is evident.

Moving forward, it will be crucial to explore interactions between stressors and personality characteristics in conjunction with specific reasons for emergency presentations. This nuanced approach may pave the way for specific characterizations of child and adolescent psychiatric emergencies, allowing for the derivation of targeted preventive measures.

It is also essential to consider the broader socioeconomic and public health contexts that existed during the SARS-CoV-2 pandemic, as these factors likely influenced family dynamics. Further qualitative exploration or additional contextual information could provide deeper insights.

This article outlines the trajectory of mental health emergencies amid the SARS-CoV-2 pandemic, recognizing it as just one of several major crises influencing children and adolescents. Beyond the SARS-CoV-2 pandemic’s direct effects, it is important to account for current and future factors, such as war and the climate crisis, as these will also shape the experiences of this population. Consequently, the SARS-CoV-2 pandemic itself was not frequently cited as the predominant aspect. Instead, individual processes appear to determine which events and stressors contribute to ideation and how they interact with personal characteristics.

### Strengths and Limitations

Using a multimethod approach, this observational study investigated the occurrences and backgrounds of child and adolescent psychiatric emergencies. While not enough is known about the characteristics of such emergencies, this study advances the field and closes some important gaps. With the help of different methodological approaches, it is possible to create a further differentiated quantification of the emergencies as well as to look at the deeper structures of the emergencies while focusing on the factors that cause stress in children and adolescents. By using a concept- and data-driven approach, the results also contribute to the theoretical framework of emergencies.

In addition to the strengths, the weaknesses of the study must also be considered. One caveat is that the study took place at only one university care center. However, because the service area was large and mixed, a diverse and broad clientele could nonetheless be considered. Still, it would be desirable to repeat the study at other CAPs. Also, we could not analyze the entire time period between the beginning of the SARS-CoV-2 pandemic and the endemic phase. Regarding the choice of time periods considered in the study, the fall and winter months were selected as phases in which a high incidence of infection could be observed, which again led to severe restrictions in the lives of children and adolescents.

In this study, it was possible to make comparisons across different periods of time. A follow-up study including a comparison with children and adolescents treated as regular outpatients or planned inpatients would certainly also be interesting, as it would allow us to differentiate the issues presented in emergency departments from other problematic situations.

## 5. Conclusions

This paper clearly shows that child and adolescent psychiatric emergencies increased as the SARS-CoV-2 pandemic progressed. This drastic development applied to the children and adolescents who were most affected by mental health problems at the given time. The acute help that is needed at such moments can be provided by child and adolescent psychiatric facilities. Therefore, child and adolescent psychiatric facilities need sufficient therapeutic and economic equipment. However, this finding can also be seen as a water-level report on the development of the mental health of the larger population of children and adolescents.

A psychiatric emergency is often preceded by a longer period of development. Within this development, plans for prevention or early interventions must be put in place in order to avoid emergencies. However, within the SARS-CoV-2-pandemic, such options were also hampered by restrictions. These restrictions applied not only to therapeutic services but also to those located in schools or recreational areas.

In any subsequent major social crises, it must be ensured that children and adolescents have access to such services despite any restrictions that are in place.

This study found that adolescents in particular (as compared with children) experienced a sharp increase in psychiatric emergencies. This empirical finding is in line with earlier reports (7,3,8) and should be taken seriously. Adolescents and their age-specific needs should now be of therapeutic, political, and social interest. In addition, the healthcare system should adapt to the aging of the generation of individuals who were children during the pandemic. It cannot be ruled out that the effects of the pandemic will have a long-term impact, and will thus lead to an increased need for treatment among those who were adolescents during the pandemic.

The study found different developments in the numbers of emergencies for different groups of disorders. Also, the study found that the reasons underlying the emergencies were heterogenous. Although there were clusters of some topics in certain periods, it turned out that the backgrounds of the emergencies were very individual. This point should be considered in emergency situations and in further treatments of the patients themselves, but also in the training of young psychiatric and psychotherapeutic professionals: Every emergency is based on very different backgrounds. Drastic restrictions in everyday life will have very different effects that depend on an individual’s personality, living conditions, and available resources. Correspondingly, any treatment and any further planning of support for children and adolescents require very individual considerations.

## Figures and Tables

**Figure 1 ijerph-21-00216-f001:**
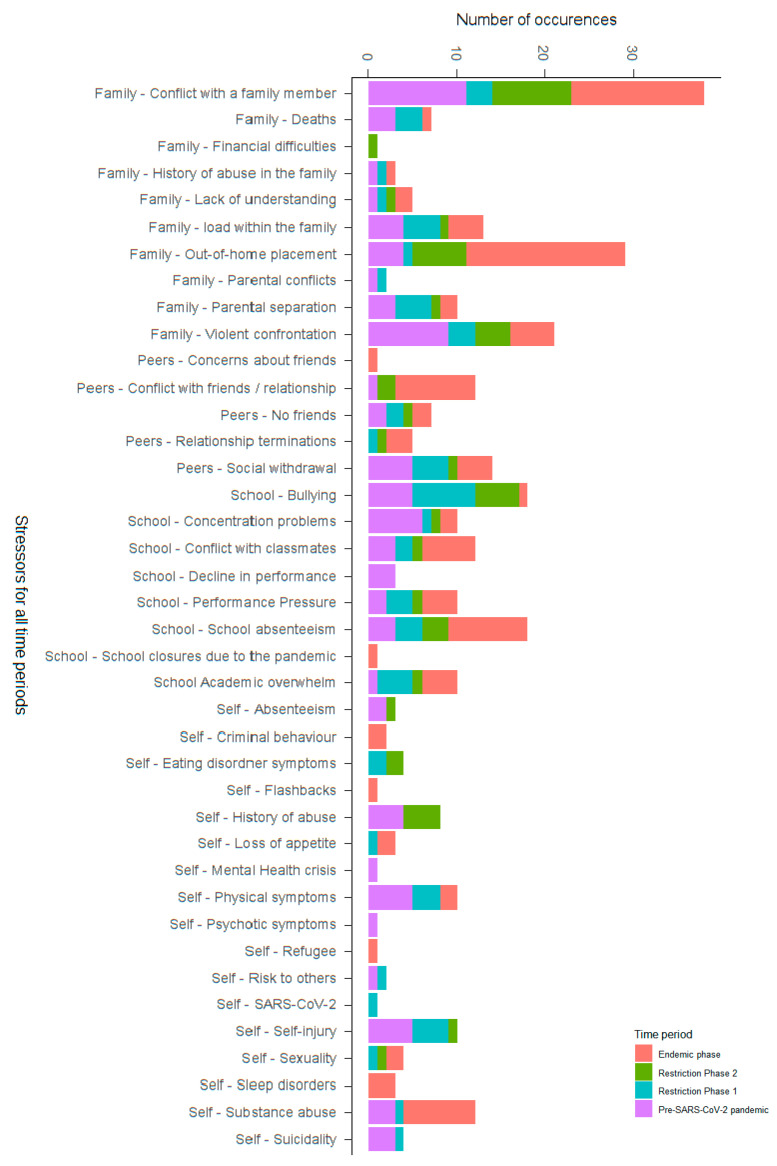
Stressors in all periods.

**Figure 2 ijerph-21-00216-f002:**
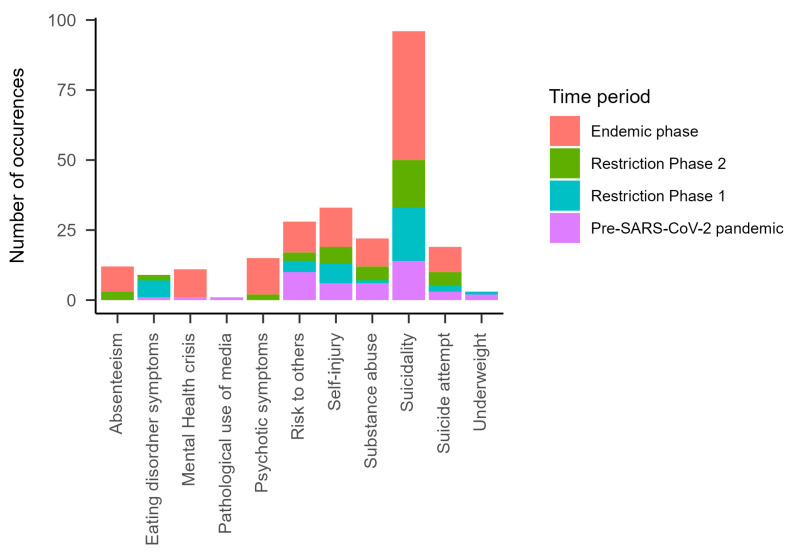
Indicators of risk/specific reasons for visits in all time periods.

**Figure 3 ijerph-21-00216-f003:**
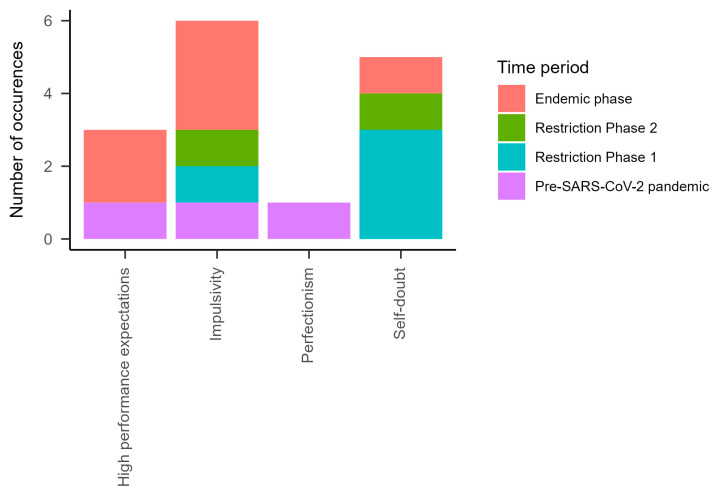
Personality characteristics in all time periods.

**Table 1 ijerph-21-00216-t001:** Criteria for selection of medical reports.

Age group	Under 13—13 patients; 13–16—41 patients; over 16—23 patients
Gender	Female—41 patients; male—36 patients No patients were stored in the system with a different gender.
Period of presentation	October 2019–January 2020—17 patients; October 20–January 21—19 patients; October 2021–January 2022—22 patients; October 22–January 23—19 patients
Weekday/Weekend	Weekday—64 patients; weekend—13 patients
Time of school year	School in session—62 patients School not in session—15 patients
Time of day	Day—51 patients; night—23 patients 3 missing values
Type of emergency presentation	Outpatients—30 patients; inpatients—47 patients
Number of emergency presentations	*M* = 4.88; *Med* = 5.50; *Min* = 1; *Max* = 20
Different medical personnel	Different physicians wrote the medical report.

**Table 3 ijerph-21-00216-t003:** Comparisons of frequency by age group for patients with different types of emergency presentations in all periods on the case level.

Kind of Emergency Presentation	Time Period	*N* Total and by Age Group ^a^						Mean Age Differences between the Groups							
		All	≤12.99 Years	13.00–15.99 Years	≥16.00 Years	χ^2^(2)	*p*	M	SD	Min	Max	F	df	*p*	^c^ η^2^
All	2019/2020	165	46 (28%)	68 (41%)	51 (31%)	18.05	0.006 *	14.15	2.75	7.12	18.32	9.14	787	0.004 *	0.01
	2020/2021	172	30 (17%)	84 (49%)	58 (34%)			14.76	2.20	6.67	17.99				
	2021/2022	222	30 (14%)	112 (50%)	80 (36%)			15.01	2.19	6.90	18.21				
	2022/2023	240	41 (17%)	99 (41%)	100 (42%)			14.92	2.54	4.71	18.65				
Telephone	2019/2020	105	37 (35%)	33 (31%)	35 (33%)	24.85	<0.001 *	13.96	2.97	7.12	18.32	7.87	616	0.009 *	0.01
	2020/2021	132	26 (20%)	65 (49%)	41 (31%)			14.63	2.26	6.67	17.99				
	2021/2022	186	26 (14%)	95 (51%)	65 (35%)			14.96	2.20	16.90	18.21				
	2022/2023	196	34 (17%)	84 (43%)	78 (40%)			14.84	2.65	4.71	18.65				
Outpatient	2019/2020	43	13 (30%)	18 (42%)	12 (28%)	7.77	0.255	13.83	2.86	7.15	17.83	1.45	268	0.229	<0.01
	2020/2021	56	8 (14%)	24 (43%)	24 (43%)			15.13	1.99	9.39	17.99				
	2021/2022	87	11 (13%)	43 (49%)	33 (38%)			14.92	2.21	8.63	17.99				
	2022/2023	84	16 (19%)	35 (42%)	33 (39%)			4.65	2.52	6.22	17.84				
Admission	2019/2020	61	7 (11%)	31 (51%)	23 (38%)	5.39	0.495	15.07	1.73	9.68	17.84	4.35	353	0.038 *	0.01
	2020/2021	78	8 (10%)	42 (54%)	28 (36%)			15.14	1.88	9.92	18.00				
	2021/2022	107	6 (6%)	59 (55%)	42 (39%)			15.47	1.16	10.54	18.00				
	2022/2023	109	8 (7%)	49 (45%)	52 (48%)			15.56	1.86	6.22	18.01				

Note. χ^2^(2) and *p* refer to chi-square independence tests of the distributions of individuals across the age groups. F, df, *p*, and η^2^ refer to ANOVAs for age group comparisons. All time periods were compared for each type of presentation. η^2^ denotes the effect size eta-squared. The first presentation of the corresponding type of presentation was considered. Telephone refers to telephone consultations, Outpatient refers to outpatient emergency visits and Admission refers to inpatient emergency admission. ^a^ The percentages refer to the total number of children and adolescents with the type of presentation and period described in the same line. ^c^ Reference values for eta-squared according to Ellis [58] were: 0.01 ≤ η^2^ < 0.06 small; 0.06 ≤ η^2^ < 0.14 medium; 0.14 ≤ η^2^ large. * describes a significant result at a significance level of α = 0.05.

**Table 4 ijerph-21-00216-t004:** Occurrences of eating disorders, obsessive–compulsive disorders, affective disorders and expansive disorders in the emergency presentations in all periods.

		Presentation Level				Case Level			
Emergency Presentation		All	Telephone	Outpatient	Admission	All	Telephone	Outpatient	Admission
Eating disorders									
*n*	all	142 (10.0%)	67 (9.7%)	24 (7.9%)	51 (12.0%)	65 (9.5%)	34 (9.7%)	20 (9.1%)	11 (9.7%)
	2019/2020	5 (2.2%)	1 (0.9%)	0	4 (5.6%)	3 (2.2%)	1 (1.6%)	0	2 (2.7%)
	2020/2021	24 (8.3%)	10 (7.4%)	6 (9.8%)	8 (8.7%)	14 (9.7%)	6 (7.9%)	5 (11.9%)	3 (11.1%)
	2021/2022	61 (13.3%)	32 (13.7%)	11 (10.8%)	18 (14.5%)	26 (12.9%)	15 (14.2%)	10 (14.7%)	1 (3.7%)
	2022/2023	52 (11.6%)	24 (10.9%)	7 (7.5%)	21 (15.3%)	22 (10.8%)	12 (11.5%)	5 (7.1%)	5 (17.4%)
	*p*	<0.001 *	<0.001 *	0.076	0.108	0.003 *	0.031 *	0.035 *	0.351
	*φ_c_*	0.09	0.10			0.08	0.09	0.13	
Obsessive–compulsive disorders									
*n*	all	38 (2.7%)	21 (3.0%)	6 (2.0%)	11 (2.6%)	17 (2.5%)	12 (3.4%)	2 (0.9%)	3 (2.7%)
	2019/2020	9 (4%)	4 (3.8%)	1 (2.1%)	4 (5.6%)	5 (3.7%)	3 (4.8%)	0	2 (6.7%)
	2020/2021	2 (0.7%)	1 (0.7%)	0	1 (1.1%)	2 (1.4%)	1 (1.3%)	0	1 (3.7%)
	2021/2022	5 (1.1%)	3 (1.3%)	1 (1.0%)	1 (0.8%)	2 (1.0%)	2 (1.9%)	0	0
	2022/2023	22 (4.9%)	13 (5.9%)	4 (4.3%)	5 (3.6%)	8 (3.9%)	6 (5.7%)	2 (2.8%)	0
	*p*	<0.001 *	0.026 *	0.234	0.125	0.155	0.330	0.333	.510
	*φ_c_*	0.08	0.08						
Affective disorders									
*n*	all	801 (56.4%)	382 (55.2%)	137 (45.1%)	282 (66.4%)	384 (45.2%)	205 (58.8%)	104 (47.1%)	75 (66.4%)
	2019/2020	115 (51.3%)	51 (49.0%)	19 (39.6%)	45 (62.5%)	66 (49.3%)	31 (49.2%)	16 (39.0%)	19 (63.3%)
	2020/2021	147 (51.0%)	67 (49.6%)	25 (41.0%)	55 (59.8%)	72 (49.7%)	40 (52.6%)	17 (40.5%)	15 (55.6%)
	2021/2022	280 (61.0%)	140 (60.1%)	52 (51.0%)	88 (71.0%)	123 (61.2%)	67 (63.2%)	37 (54.4%)	19 (70.4%)
	2022/2023	259 (57.6%)	124 (56.4%)	41 (44.1%)	94 (68.6%)	123 (60.6%)	67 (64.4%)	34 (48.6%)	22 (75.9%)
	*p*	0.019 *	0.129	0.484	0.288	0.033 *	0.124	0.351	0.417
	*φ_c_*	0.06				0.08			
Expansive disorders									
*n*	all	216 (15.2%)	116 (16.8%)	49 (16.1%)	51 (12.0%)	114 (16.7%)	61 (17.5%)	34 (15.4%)	19 (16.8%)
	2019/2020	55 (24.6%)	31 (29.8%)	12 (25.0%)	12 (16.7%)	34 (25.4%)	20 (31.7%)	9 (22.0%)	5 (16.7%)
	2020/2021	60 (20.8%)	29 (21.5%)	12 (19.7%)	19 (20.7%)	32 (22.1%)	18 (23.7%)	9 (21.4%)	5 (18.5%)
	2021/2022	54 (11.8%)	29 (12.4%)	17 (16.7%)	8 (6.5%)	22 (10.9%)	9 (8.5%)	9 (13.2%)	4 (14.8%)
	2022/2023	47(10.4%)	27 (12.3%)	8 (8.6%)	12 (8.8%)	26 (12.8%)	14 (13.5%)	7 (10.0%)	5 (17.2%)
	*p*	<0.001 *	<0.001 *	0.052	0.005*	<0.001 *	<0.001 *	0.215	1.000
	*φ_c_*	0.11	0.12		0.12	0.11	0.14		

Note. *n* refers to sample size. *p* and *φ_c_* (Cramer’s V effect size) refer to Fisher’s exact tests, which were used to investigate the independence of the proportions of the respective clinical profiles from the observation periods. * describes a significant result at a significance level of α = 0.05. Telephone refers to telephone consultations, Outpatient refers to outpatient emergency visits and Admission refers to inpatient emergency admission.

**Table 5 ijerph-21-00216-t005:** Occurrences of anxiety disorders, substance abuse and psychoses in the emergency presentations in all periods.

		Presentation Level				Case Level			
Emergency Presentation		All	Telephone	Outpatient	Admission	All	Telephone	Outpatient	Admission
Anxiety disorders									
*n*	all	159 (11.2%)	84 (12.1%)	31 (10.2%)	44 (10.4%)	84 (12.3%)	51 (14.6%)	20 (9.0%)	13 (11.5%)
	2019/2020	16 (7.1%)	9 (9.7%)	3 (6.3%)	4 (5.6%)	10 (7.5%)	7 (11.1%)	1 (2.4%)	2 (6.7%)
	2020/2021	30 (10.4%)	14 (10.4%)	6 (9.8%)	10 (10.9%)	14 (9.7%)	7 (9.2%)	4 (9.5%)	3 (11.1%)
	2021/2022	54 (11.8%)	31 (13.3%)	8 (7.8%)	15 (12.1%)	29 (14.4%)	21 (19.8%)	4 (5.9%)	4 (14.8%)
	2022/2023	59 (13.1%)	30 (13.6%)	13 (15.1%)	15 (10.9%)	31 (15.3%)	16 (15.4%)	11 (15.7%)	4 (13.8%)
	*p*	0.120	0.527	0.312	0.515	0.092	0.209	0.093	0.761
	*φ_c_*								
Substance abuse									
*n*	all	103 (7.2%)	46 (6.6%)	29 (9.5%)	29 (6.6%)	47 (6.9%)	17 (4.9%)	20 (9.0%)	10 (8.8%)
	2019/2020	19 (8.5%)	6 (5.8%)	4 (8.3%)	9 (12.5%)	14 (10.4%)	4 (6.3%)	4 (9.8%)	6 (20.0%)
	2020/2021	26 (9.0%)	12 (8.9%)	9 (14.8%)	5 (5.4%)	15 (10.3%)	5 (6.6%)	8 (19.0%)	2 (7.4%)
	2021/2022	26 (5.7%)	14 (6.0%)	8 (7.8%)	4 (3.2%)	10 (5.0%)	5 (4.7%)	5 (7.4%)	0
	2022/2023	32 (7.1%)	14 (6.4%)	8 (8.6%)	10 (7.3%)	8 (3.9%)	3 (2.9%)	3 (4.3%)	2 (6.9%)
	*p*	0.290	0.718	0.503	0.089	0.025 *	0.615	0.079	0.057
	*φ_c_*					0.08			
Psychoses									
*n*	all	33 (2.3%)	12 (1.7%)	9 (3.0%)	12 (2.8%)	12 (1.8%)	4 (1.1%)	5 (2.3%)	3 (2.7%)
	2019/2020	3 (1.3%)	0	2 (4.2%)	1 (1.4%)	2 (1.5%)	0	1 (2.4%)	1 (3.3%)
	2020/2021	10 (3.5%)	3 (2.2%)	3 (4.9%)	4 (4.3%)	4 (2.8%)	0	2 (4.8%)	2 (7.4%)
	2021/2022	7 (1.5%)	2 (0.9%)	2 (2.0%)	3 (2.4%)	3 (1.5%)	1 (0.9%)	2 (2.9%)	0
	2022/2023	13 (2.9%)	7 (3.25)	2 (2.1%)	4 (2.9%)	3 (1.5%)	3 (2.9%)	0	0
	*p*	0.214	0.128	0.630	0.751	0.831	0.395	0.264	0.316
	*φ_c_*								

Note. *n* refers to sample size. *p* and *φ_c_* (Cramer’s V effect size) refer to Fisher’s exact tests, which were used to investigate the independence of the proportions of the respective clinical profiles from the observation periods. * describes a significant result at a significance level of α = 0.05. Telephone refers to telephone consultations, Outpatient refers to outpatient emergency visits and Admission refers to inpatient emergency admission.

**Table 6 ijerph-21-00216-t006:** Hierarchical category system representing the reasons for emergency visits.

Indicators of Risk/Specific Reason for Visit	Stressors				Personality Characteristics
	School	Peers	Family	Self	
Underweight	School closures due to the pandemic	Concerns about friends	Out-of-home placement	Psychotic symptoms	High performance expectations
Eating disorder symptoms	School absenteeism	Relationship terminations	Deaths	Risk to others	Perfectionism
Pathological use of media	Academic overwhelm	Conflict with friends/relationship	Financial difficulties	Mental health crisis	Impulsivity
Substance abuse	Performance pressure	No friends	Violent confrontation	Absenteeism	Self-doubt
Absenteeism	Decline in performance	Social withdrawal	History of abuse in the family	Eating disorder symptoms	
Mental health crisis	Concentration problems		Mental disorder, deviant behavior, or disability within the family	Suicidality	
Self-injury	Conflict with classmates		Parental separation	Self-injury	
Psychotic symptoms	Bullying		Parental conflicts	Refugee	
Risk to others			Parent´s lack of understanding towards the children	Criminal behavior	
Suicide attempt			Conflict with family member/caregiver	Flashbacks	
Suicidality				Loss of appetite	
Underweight				Sleep disorders	
				Substance abuse	
				SARS-CoV-2 pandemic	
				Physical symptoms	

## Data Availability

The raw data supporting the conclusions of this article will be made available by the authors on request.

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
