# Peer review of "SARS-CoV-2 and Adolescent Psychiatric Emergencies at the Tübingen University Hospital: Analyzing Trends, Diagnoses, and Contributing Factors"

_ijerph, 2024, doi:10.3390/ijerph21020216_

Round 1

Reviewer 1 Report

Comments and Suggestions for Authors

·         This paper aims to analyze the changes in adolescent psychiatric emergencies due to the SARS CoV2-19 pandemic. The authors expose data regarding reasons for demanding emergency attention, as well as stress factors and personality traits, (this latter based on patients’ background). The sample is made of hospital records from adolescent patients in a University hospital. All of them have been diagnosed as having either eating disorders or obsessive-compulsive disorders (OCD) (I am not sure of this information, please check the criteria inclusion and try to be concise and clear). Here are some suggestions that I hope will help you to revise the manuscript

·         125 better transition

·         262-263 which is the n in this case (all relevant patients). Also, which are relevant patients? This should be explained.

·         264- why only 77 patients’ data are used for questions 2a and 2b

·         297—the patients were distributed in different groups to analyze differences in their age? In how many?

·         301 why it is important to see the differences just in ED and OCD? Not clear

·         338 it seems to be double space before “In this way”.

·         330-331 please re-write to add clarity.

·         361—“case level”. I think it is necessary to explain what do you mean (you say in page 6 “each person was only considered once per period”--- so, there are 2 units of analysis: each medical report / intervention, and each patient. Maybe you could state this in a clearer way to help the reader understand the data.

·         378. This is the analysis of the presentations? Because in line 372 says that there is a significant different in age at the conceptual level. Line 382---still talking about the age? Not clear. This paragraph is a bit confusing.

·         Lune 413—I think would be important to explain why eating disorder diagnosis is so important in this analysis, as well as the OCD (we can see in the results that OCD is a frequent stressor, but this should be explained in the introduction, in my opinion). I also consider that in the introduction there is a lot of information about suicidal conduct, but then in the results, nothing is said about this. So why explain all that in the intro? 

I would add some information about the impact of these results (on policies, on service delivery, on training future professionals...). I think this could amplify the interest of the readers on this study.

Comments on the Quality of English Language

Check some large sentences to make it more readible

Reviewer 2 Report

Comments and Suggestions for Authors

SARS-CoV-2-19, SARS CoV2-19, Sars-CoV2-19 – why alternatively spelled? ‘Mental stress can lead to states of emergency that necessitate an emergency presentation. emergency presentations’ – poorly constructed, also avoid close word repetitions, and why not starting the latter sentence with initial capitalization? ‘so-called emergency presentations in child and adolescent psychiatric units’ – why ‘so-called’? ‘from the 2021 examination and to examine’ – avoid close word repetitions. You somehow alternatively use children, adolescents, students, and young people. Your in-text citation style is variable. You sometimes use names + number, other times only the number. Lauwerie et al. [14] described that more girls were admitted as emergencies during the SARS CoV2-19-pandemic than before. Furthermore, they were able to show that more youths with no history of mental health or suicide encounters were admitted to emergency departments in suicidal crisis than before the SARS CoV2-19-pandemic. Other studies in the US described an increase in suicide attempts among adolescents (26,2% to 50,6% increase) presentatios in emergency departments [14]. – You start with Lauwerie et al. [14] and end with [14], but in the last sentence you write ‘Other studies in the US…’ without citing them. ‘Although completed suicides in children and adolescents are rare, they are still the second most common cause of death after accidental death [16].’ – how are they rare if they are ‘the second’? ‘While, the suicide rate in Germany’ – remove the comma and add it after ‘1980s’. Some acronyms have not been written first in full. ‘Respective reviews overall describe’ – what reviews? ‘The responses of OCD-patients are mixed with studies finding a deterioration despite treatment and single studies finding no worsening of symptoms at the beginning of the SARS CoV2-19-pandemic [23]’ – ‘studies’, but you cite a single source. ‘These demands need to be dealt with and a lack of sufficient coping strategies can cause significant stress. Failure to cope with this stress can, under certain circumstances, lead to the development of mental illness or behavioral problems [10]. Development is linked to biological changes and the demands of the social context, and is also subject to cultural influences [10].’ – why mentioning the same source 2 times in a row? ‘Ougrin and colleagues’, ‘Steinhoff and colleagues’ – for style consistency use ‘et al.’. ‘Steinhoff and colleagues [28] identify schools, peers, intimate relationships, and family as social contexts in which stressful life events can occur. The events in schools include grade repetition and failing exams. Here, the peers context includes violence and sexual victimization, intimate relationships include, the loss of friends and separation from partners, and the family context includes loss and instability. Academic stress factors, such as perfectionism or truancy, as well as social exclusion, are also associated with a higher risk of self-harming behavior [29].’ – where does 28 end and 29 starts? Check for no or extra spaces throughout the manuscript. E.g., ‘illnesses. However’. ‘For the adolescents’ – remove ‘the’. ‘it was challenging that during the lockdown phases’ – poorly constructed. ‘It's essential’ – avoid word abbreviations. ‘External stressors, such as the SARS-CoV-2-19 pandemic, have had an impact on mental health.’ – poorly constructed. ‘Culminating stressors in different areas of life led to psychological crises in children and adolescents.’ – have led.
The relationship between COVID-19-related viral panic and contagious fear as regards SARS-CoV-2-based adolescent psychiatric emergencies has not been covered, and thus such sources can be cited:
Lăzăroiu, G., Horak, J., and Valaskova, K. (2020). “Scaring Ourselves to Death in the Time of COVID-19: Pandemic Awareness, Virus Anxiety, and Contagious Fear,” Linguistic and Philosophical Investigations 19: 114–120. doi: 10.22381/LPI1920208
Lăzăroiu, G., and Adams, C. (2020). “Viral Panic and Contagious Fear in Scary Times: The Proliferation of COVID-19 Misinformation and Fake News,” Analysis and Metaphysics 19: 80–86. doi:10.22381/AM1920209
Bratu, S. (2020). “The Fake News Sociology of COVID-19 Pandemic Fear: Dangerously Inaccurate Beliefs, Emotional Contagion, and Conspiracy Ideation,” Linguistic and Philosophical Investigations 19: 128–134. doi: 10.22381/LPI19202010

Comments on the Quality of English Language

SARS-CoV-2-19, SARS CoV2-19, Sars-CoV2-19 – why alternatively spelled? ‘Mental stress can lead to states of emergency that necessitate an emergency presentation. emergency presentations’ – poorly constructed, also avoid close word repetitions, and why not starting the latter sentence with initial capitalization? ‘so-called emergency presentations in child and adolescent psychiatric units’ – why ‘so-called’? ‘from the 2021 examination and to examine’ – avoid close word repetitions. You somehow alternatively use children, adolescents, students, and young people. ‘While, the suicide rate in Germany’ – remove the comma and add it after ‘1980s’. ‘For the adolescents’ – remove ‘the’. ‘it was challenging that during the lockdown phases’ – poorly constructed. ‘It's essential’ – avoid word abbreviations. ‘External stressors, such as the SARS-CoV-2-19 pandemic, have had an impact on mental health.’ – poorly constructed. ‘Culminating stressors in different areas of life led to psychological crises in children and adolescents.’ – have led.

Reviewer 3 Report

Comments and Suggestions for Authors

The study with the title “SARS-CoV-2 and Adolescent Psychiatric Emergencies at the Tübingen University Hospital: analyzing trends, diagnoses, and contributing factors“ (Manuscript ID: ijerph-2776588) aims to investigate the changes in the number of psychiatric emergencies, specifically focusing on the impact of the SARS-CoV-2-19 pandemic at Child and Adolescent Psychiatry (CAP), Tübingen. It considers age, eating disorders, and obsessive-compulsive disorders (OCD) in updating a previous study, employing a mixed-method approach to assess both quantitative and qualitative aspects of emergency contacts. The study also evaluates the thematic background of emergencies and, in light of the increasing numbers, emphasizes the long-term goal for the medical community to meet growing demands and establish preventive options.

This study explores a crucial subject and aligns with the objectives of the special issue, "Mental Health of Children and Adolescents: Tackling the Consequences of Major Crises."

However, I have following major and minor concerns:

Comments on the Abstract:

Lines 21 and 22: The terms "lockdown A" and "lockdown B" should be supplemented with time references to provide clarity about the respective time periods they represent.

Line 24: The abbreviation “OCD” appears for the first time in the text and should be spelled out.

Comments on the Introduction section:

The introduction requires a comprehensive revision as there is a lack of a clear structure, and information (especially regarding previous research) is presented in a seemingly random and volatile manner. It remains unclear why EDs and OCDs were specifically chosen for investigation. While the original study by Allgaier et al. (2022) provides justification, this rationale is absent in the current study. Merely stating that the current study is an update of the previous one does not sufficiently explain the choice of focusing on EDs and OCDs. A more detailed and explicit rationale is needed to enhance the coherence and clarity of the introduction.

Lines 58 and 59: I assume that the terms (“lockdown A” and “B”) were chosen by the authors; please mark them accordingly. Specify whether the lockdowns occurred solely in Tübingen or nationwide in Germany.

Lines 82 and 83: Specify the time period that was investigated in the study by Allgaier et al. (2022).

Line 114: The abbreviation “ED” appears for the first time in the text and should be spelled out as “eating disorders”.

Lines 128-130: Franzen et al. (2020) examined NSSI, while Ougrin et al. (2020) investigated self-harm. It is not permissible to conflate these two terms under the umbrella of "non-suicidal self-harm (NSSH)”. Please use either "self-harm", as it encompasses both terms, or explicitly use both "self-harm" and "non-suicidal self-injury" to maintain clarity and accuracy. The provided information regarding 50% of emergency presentations involving self-harm is not found in the cited reference by Ougrin et al. (2020). It would be appropriate to include the following article at this point: Wong, B. H. C., Cross, S., Zavaleta-Ramírez, P., Bauda, I., Hoffman, P., Ibeziako, P., ... & Ougrin, D. (2023). "Self-harm in children and adolescents who presented at emergency units during the COVID-19 pandemic: An international retrospective cohort study." Journal of the American Academy of Child & Adolescent Psychiatry.

Comments on the Materials and Methods section:

An explanation for the choice of specific age groups (under 13 years, 13-16 years, etc.) would be necessary.

Lines 245 and 246: It is not clearly specified whether "emergency presentation" is an overarching term encompassing "emergency admission", "outpatient emergency visit", and "emergency telephone consultation", or if it represents one of these subcategories, or both.

Line 261: The information for all assessed categories regarding sex assigned at birth and/or gender is not explicitly provided or detailed. Please provide additional information.

Comments on the Results section:

The results should be visualized more effectively and comprehensibly, especially in Figures 5, which appears cluttered and unclear.

Lines 486-500: It should be clearly stated that all subcategories belong to the second main category. While this is evident in Table 5, it should also be explicitly reported in the text. Additionally, the distinction between the subcategory "self" and the main category "personality characteristics" should be better emphasized.

Comments on the Discussion section:

The Discussion section is well-written and understandable, offering a clear synthesis of the study's findings and their broader implications in the context of child and adolescent psychiatric emergencies during the SARS-CoV-2-19 pandemic.

Lines 631 and 649: Citations specifying the previous studies referred to should be added.

The role of the use of social media, as it also occurs as a specific reason for visits, could be discussed. In this context, the following study could also be relevant: Laczkovics, C., Lozar, A., Bock, M. M., Reichmann, A., Pfeffer, B., Bauda, I., ... & Kothgassner, O. (2023). Psychische Gesundheit und Social Media-Nutzungsverhalten von Jugendlichen und jungen Erwachsenen während der COVID-19-Pandemie. Praxis der Kinderpsychologie und Kinderpsychiatrie, 72(7), 591-604.

Thank you for the opportunity to review this manuscript, and I wish the authors success in their future work.

Comments on the Quality of English Language

Generally, there are numerous typos, repetitions of words and the verb tenses often do not align with the presented content. In the Results section, there is complete chaos regarding which statistical parameters are italicized and not.

Here some examples:

Line 15: There are several variations of SARS-CoV-2-19 (e.g., SARS CoV2-19 or SARS-CoV2-19). Please unify them.

Line 43: A new sentence should begin with a capital letter.

Line 62: The word "implemented" does not fit the context, and there is a question about from whom the developmental tasks should be "implemented."

Line 104: "Presentatios" instead of "presentations."

Lines 82, 158, 162: Past tense should be used.

Round 2

Reviewer 2 Report

Comments and Suggestions for Authors

This revised version can be published.

Reviewer 3 Report

Comments and Suggestions for Authors

Thank you for the thorough revision of your manuscript. I appreciate the effort and attention to detail you have invested in addressing the previously highlighted concerns.

The revisions have significantly enhanced the clarity and depth of the paper. From my point of view, your paper can be accepted for publication.